# Associations between HIV status and self-reported hypertension in a high HIV prevalence sub-Saharan African population: a cross-sectional study

Katherine Davis ,[1] Louisa Moorhouse,[1] Rufurwokuda Maswera,[2] Phyllis Mandizvidza,[2] Tawanda Dadirai,[2] Tafadzwa Museka,[2] Constance Nyamukapa,[1,2] Mikaela Smit,[1] Simon Gregson[1]

[1]MRC Centre for Global Infectious Disease Analysis, Department of Infectious Disease Epidemiology, Imperial College London, London, UK
[2]Biomedical Research and Training Institute, Harare, Zimbabwe

**Correspondence to**
Ms Katherine Davis;
K.davis18@imperial.ac.uk

## ABSTRACT

**Objectives** This study examined whether HIV status and antiretroviral therapy (ART) exposure were associated with self-reported hypertension in Zimbabwe.

**Design** Study data were taken from a cross-sectional, general population survey, which included HIV testing (July 2018–December 2019).

**Setting** The data were collected in Manicaland Province, Zimbabwe.

**Participants** 9780 people aged 15 years and above were included.

**Outcome measure** Self-reported hypertension was the outcome measure. This was defined as reporting a previous diagnosis of hypertension by a doctor or nurse. After weighting of survey responses by age and sex using household census data, $\chi^2$ tests and logistic regression were used to explore whether HIV status and ART exposure were associated with self-reported hypertension.

**Results** The weighted prevalence of self-reported hypertension was 13.6% (95% CI 12.9% to 14.2%) and the weighted prevalence of HIV was 11.1% (10.4% to 11.7%). In univariable analyses, there was no evidence of a difference in the weighted prevalence of self-reported hypertension between people living with HIV (PLHIV) and HIV-negative people (14.1%, 11.9% to 16.3% vs 13.3%, 12.6% to 14.0%; p=0.503) or between ART-exposed and ART-naive PLHIV (14.8%, 12.0% to 17.7% vs 12.8%, 9.1% to 16.4%,p=0.388). Adjusting for socio-demographic variables in logistic regression did not alter this finding (ORs:HIV status:0.88, 0.70 to 1.10, p=0.261; ART exposure:0.83, 0.53 to 1.30, p=0.411).

**Conclusions** Approximately one in seven PLHIV self-reported having hypertension, highlighting an important burden of disease. However, no associations were found between HIV status or ART exposure and self-reported hypertension, suggesting that it will be valuable to focus on managing other risk factors for hypertension in this population. These findings should be fully accounted for as Zimbabwe reorients its health system towards non-communicable disease control and management.

## STRENGTHS AND LIMITATIONS OF THIS STUDY

⇒ This is the first study to compare the prevalence of self-reported hypertension between people living with HIV and HIV-negative people in a non-clinical setting in Zimbabwe.

⇒ We used data from a large population survey, which included a diverse range of men and women, and could be weighted to local census information, increasing the representativeness of our results.

⇒ We focused on self-reported hypertension, so hypertensive people who were unaware of their status were not included in our prevalence estimates.

⇒ Our study used cross-sectional data that excluded individuals in hospital settings and did not capture data on all risk factors for hypertension.

## INTRODUCTION

Hypertension is the leading modifiable cause of death and disability worldwide.[1] In sub-Saharan Africa, the region most affected by infectious diseases such as HIV, the burden of hypertension is growing rapidly.[2] Yet despite these newly coinciding burdens, the relationship between HIV and hypertension is poorly understood.[3]

Various plausible mechanisms that could generate a difference in the prevalence of hypertension between people living with HIV (PLHIV) and HIV-negative people have been proposed. For example, chronic inflammation, increased microbial translocation, renal disease and blood vessel damage resulting from long-term ART exposure could increase the prevalence of hypertension among PLHIV, as could higher levels of behavioural risk factors among PLHIV in some communities.[4–6] Conversely, possible reasons for a reduced hypertension burden among PLHIV include low blood pressure resulting from advanced HIV disease, better control of blood pressure due to additional healthcare support and lower levels of behavioural risk factors among PLHIV in some settings.[7–9] The conflicting outcomes of the proposed

mechanisms mean that it is unclear whether any difference in hypertension prevalence by HIV status exists and in what direction it acts.

As a variety of physiological and treatment-related mechanisms by which HIV could raise or lower hypertension risk have been proposed, epidemiological studies are vital to furthering understanding.[5 7] Globally, there is some observational evidence that PLHIV might face a higher burden of hypertension.[4 6 10–12] However, the relationship appears to vary by region and it has been suggested that, in African countries, PLHIV may experience a lower prevalence of hypertension.[6] For example, in a Ugandan population, PLHIV were found to have 30% lower odds of hypertension than matched HIV-negative controls.[13] Detailed information about how the burden of hypertension differs by HIV status remains scarce across sub-Saharan Africa, constraining understanding of the association between HIV and hypertension; this is of particular concern as the relationship may differ between populations.[6] In Zimbabwe, only three studies have examined the burden of hypertension among PLHIV in the community, and all lacked a comparison group of HIV-negative people.[14–16] Just one of these studies examined the effect of antiretroviral therapy (ART) on hypertension, reporting that long-term use of ART was associated with increased prevalence of hypertension, a finding that has not been replicated in this setting.[14]

Bridging the data gap will be crucial to supporting strategic decision-making about provision of preventative services and hypertension treatment. Currently, capacity to prevent and control hypertension in Zimbabwe is limited, but major changes are being planned to manage the rising burden.[14 17–19] Delivering these changes efficiently will require an understanding of the role of HIV status and ART exposure in hypertension. This study aimed to contribute to that understanding, by examining whether HIV status and ART exposure were associated with self-reported hypertension in Manicaland, East Zimbabwe.

## METHODS

### Data

Study data were taken from the Manicaland General Population Open Cohort Study (Manicaland Study), a long-running, large-scale HIV sero-survey.[20 21] Each survey round of the cohort has involved testing participants for HIV and asking questions on topics including participants' sociodemographic characteristics, general health and knowledge of HIV.[21] The most recent round, which ran between July 2018 and December 2019, is unique as it provides the first insight into the burden of hypertension and other non-communicable diseases. Consequently, cross-sectional data from the most recent round were used in this analysis (questionnaire in online supplemental material).

Eight sites were included in the round: a subsistence farming area (Bonda Mission), a tea estate (Eastern Highlands), a forestry estate (Selbourne), two small towns (Nyazura and Nyanga), a roadside settlement (Watsomba) and two urban areas (Sakubva and Hobhouse). A team of Zimbabwean researchers led data collection at the sites, gathering survey responses on electronic tablets. Initially, households were enumerated in a census that included a brief questionnaire that asked about relationships between those in the household and household assets. After the census, more detailed data were collected through individual interviews. All women aged 15–24 years and all men aged 15–29 years, as well as a random sample of two thirds of older adults, in enumerated households, were eligible to complete the individual interview. The research team contacted eligible participants and interviewed them at central sites, such as health centres, or in their homes. If eligible people were unavailable or could not be contacted when first approached, at least two further attempts were made to reach them. The team aimed to achieve a sample size of approximately 9500, in order to detect changes in the primary outcome measures of the Manicaland Study, which will be reported elsewhere.[22–24]

During the individual interview, participants provided a range of sociodemographic and health information. For example, they self-reported whether they had ever been diagnosed with hypertension by a doctor or nurse and whether they had ever taken ART.

In addition to responding to interview questions, participants were offered provider-initiated testing and counselling (PITC) for HIV and asked to provide a dried blood spot (DBS) specimen for laboratory HIV testing. PITC was performed by accredited members of the research team following the standard procedures and testing algorithm used in Ministry of Health and Child Care routine PITC services. DBS specimens were tested using the same algorithm at the Biomedical Research and Training Institute in Harare.

Funding limitations meant that blood pressure was not measured during the household census or individual interview.

### Weighting

The selection criteria for the individual interview, and the pattern of response and non-response, generated a young cohort, with more women than men. As a result, weights were used to adjust variable distributions to better match the population in the household census. First, census data were analysed to calculate the proportion of the population in 5-year age groups between 15 and 74 years, as well as a 75+-year-old age group, by sex. Iterative proportional fitting, also known as raking, was then used to generate weights that matched the age and sex proportions in the interview sample to those in the household census. Participant data were weighted for all further analyses; this approach meant that 42 people with missing age or sex information were excluded.

### Association between HIV and self-reported hypertension

After weighing, prevalence estimates for self-reported hypertension and HIV were generated. Self-reported

hypertension was defined as reporting ever having been diagnosed with hypertension by a doctor or nurse. HIV status was determined from PITC results; however, if a person opted out of PITC but consented to provide a DBS specimen, the DBS specimen was tested in the laboratory and this result was used instead. Consequently, our sample of PLHIV included both people who knew their status at the time of interview and people who did not. Prevalence estimates were calculated as the weighted number of PLHIV or self-reporting hypertension, divided by the weighted total population. To compare the prevalence of self-reported hypertension between PLHIV and HIV-negative people, estimates for the prevalence of self-reported hypertension stratified by HIV status, age and sex were produced in the same manner. Age was grouped into six classes: '15–24', '25–34', '35–44', '45–54', '55–64' '65 and over'. Categories were chosen to reflect the Zimbabwean population distribution, in which most people are young, but a minority have reached old age. Crude comparisons of the prevalence of self-reported hypertension between PLHIV and HIV-negative people were made using $X^2$ tests, with Rao-Scott adjustment. These were performed both for the overall population and within age and sex strata.

Logistic regression models were employed to examine the relationship between HIV and self-reported hypertension, while controlling for available sociodemographic variables. All models were complete case analyses. Initially, a model was built including HIV, age and sex. This was then developed into a fully adjusted model. First, a range of potential sociodemographic confounders were identified through the literature. Second, any potential confounders that were found to be associated with self-reported hypertension in models including age and sex, at a threshold p value of 0.1, were added to the initial model. Lastly, the new model was refined by backward elimination of potential confounders, with a threshold p value of 0.05. This created a final model for generating adjusted estimates of the association between HIV status and self-reported hypertension.

The potential confounders that were considered for inclusion in the model were employment status, relationship status, education status, study site and wealth. Employment status was classified as 'employed or in education' or 'unemployed'; people who worked in the informal sector, doing petty trading or agricultural work, were categorised as 'employed or in education'. Relationship status was coded as 'married or in a long-term relationship', 'never been in a long-term relationship', 'separated or divorced' or 'widowed'. A long-term relationship was defined as a relationship of 12 months or more. Assessment of education status was based on the highest level of schooling that a person had obtained. Two categories were used: 'none or primary' and 'secondary or higher'; this reflected the fact that few people reported no schooling or education above secondary level. Wealth was classified using an index based on whether households owned a range of assets, as described by Lopman

*et al.*[25] Assets included sellable items, such as bicycles and cattle, as well as fixed assets, such as water and electricity supplies. For each asset, binary and categorical responses were coded so that they lay between zero and one. For example, owning a bicycle was coded as one, whereas not owning a bicycle was coded as zero. The values for each asset were then summed, and the total was divided by the overall number of assets, to give an asset ownership score between zero (no assets owned) and one (all assets owned) for each household. Households were then grouped into five equally sized quintiles to create a household level wealth index with five categories. Each person in the dataset was then given the wealth index for their household.

The category representing the absence of a characteristic was chosen as the baseline for most variables. For variables where there was not an absent category, the category with the most participants, or that had been chosen as the baseline in other analyses of Manicaland Study data, was selected.[26 27]

## Association between ART-exposure and self-reported hypertension

The association between ART exposure in PLHIV and self-reported hypertension was evaluated using the approach to logistic regression described above. People who reported ever taking ART were defined as ART-exposed, allowing the prevalence of ART exposure to be calculated as the weighted number of PLHIV who were ART-exposed, divided by the weighted total number of PLHIV. Prevalence estimates for self-reported hypertension, stratified by ART exposure, age, and sex, were then produced and a crude comparison between ART-exposed and ART-naive PLHIV was made using $X^2$ tests, with Rao-Scott adjustment. Finally, logistic regression models were developed, using the same model building strategy as was applied for the HIV status analysis, to produce adjusted estimates.

As a sensitivity analysis, we explored how the results changed when current ART usage, defined as reporting both having ever taken ART and not having stopped taking ART, was assessed instead of ART exposure, using the approach described above. We also used the same methods to examine the association between adherent current ART usage and hypertension, with adherent current ART usage defined as reporting all the following: having ever taken ART, not having stopped taking ART, taking ART all the time, not forgetting to take ART and having taken ART every day in the last month.

## Software
Analyses were performed in R V.4.0.3 using the survey package.[28 29]

## Patient and public involvement
Members of the local community provided input into the research through small meetings held with community members at the beginning of the Manicaland Study and

through providing feedback at meetings that were held to disseminate results in successive rounds of the survey. Feedback included recommendations for improving the recruitment and conduct of the study, as well as for reducing the burden of participation. In addition, the team worked closely with local guides (mainly community health workers) who also provided feedback as the study progressed. Input into the detail of the study design, research questions and outcome measures, was provided by local members of the research team too.

It is intended that the main results of this study will be disseminated to local stakeholders, research participants and other community members, as part of a wider plan to share results from the Manicaland Study. As in previous survey rounds, involvement of community members and local members of the research team will be key to the development of the dissemination strategy.

## RESULTS

During the census, 7610 households were contacted (figure 1). Of these, 7550 households (99.2%) completed the questionnaire, making some of their members eligible for the individual interview. Among the 12 761 eligible people, 2939 (23.0%) were unable to participate in, or did not consent to, the individual interview (reasons shown in figure 1). More of those who did not consent came from wealthier households ($\chi^2$=47.9, p<0.001) and urban sites ($\chi^2$=158.89, p<0.001). After exclusion of people without age or sex data, there were 9780 participants in the study population.

Following weighting, there were marginally more women than men in the interviewed population (women: 52.2%; table 1). The population was young; nearly a third were under 25 years old (31.0%). Unemployment was

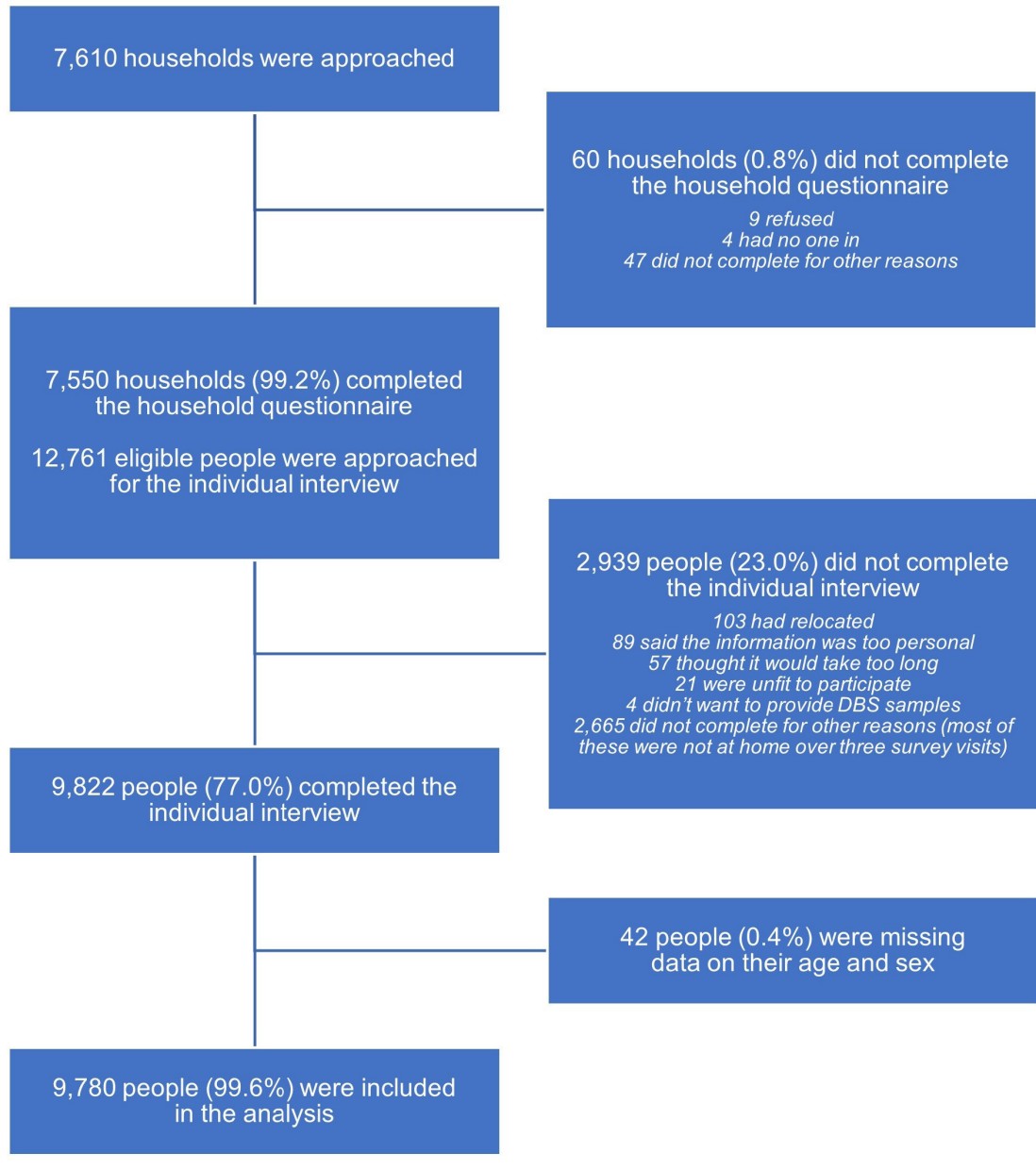

**Figure 1** Flow of study participants. DBS, dried blood spot.

**Table 1** Characteristics of study participants

| | Total population (N=9780)* | | People living with HIV (N=936)* | | HIV-negative people (N=8378)* | |
|---|---|---|---|---|---|---|
| | Weighted proportion, % (95% CI) | N* | Weighted proportion, % (95% CI) | N* | Weighted proportion, % (95% CI) | N* |
| **Hypertension** | | | | | | |
| Not reported | 86.4% (85.7% to 87.1%) | 8453 | 85.9% (83.7% to 88.1%) | 791 | 86.7% (86.0% to 87.4%) | 7273 |
| Reported | 13.6% (12.9% to 14.2%) | 1252 | 14.1% (11.9% to 16.3%) | 139 | 13.3% (12.6% to 14.0%) | 1043 |
| Missing | – | 75 | – | 6 | – | 62 |
| **HIV status** | | | | | | |
| Negative | 88.9% (88.3% to 89.6%) | 8378 | – | – | – | – |
| Positive | 11.1% (10.4% to 11.7%) | 936 | – | – | – | – |
| Missing | – | 466 | – | – | – | – |
| **ART exposure** | | | | | | |
| ART-naïve | – | – | 34.1% (31.0% to 37.2%) | 323 | – | – |
| ART-exposed | – | – | 65.8% (62.8% to 69.0%) | 612 | – | – |
| Missing | – | – | – | 1 | – | – |
| **Sex†** | | | | | | |
| Male | 47.8% | 4064 | 43.6% (40.6% to 46.7%) | 331 | 48.2% (47.8% to 48.7%) | 3544 |
| Female | 52.2% | 5716 | 56.4% (53.3% to 59.4%) | 605 | 51.8% (51.3% to 52.2%) | 4834 |
| **Age, years†** | | | | | | |
| 15–24 | 31.0% | 3923 | 7.1% (5.8% to 8.5%) | 97 | 34.7% (34.4% to 35.1%) | 3708 |
| 25–34 | 22.0% | 1988 | 19.8% (17.4% to 22.2%) | 191 | 22.2% (21.9% to 22.6%) | 1689 |
| 35–44 | 18.5% | 1561 | 31.4% (28.6% to 34.2%) | 285 | 16.9% (16.5% to 17.3%) | 1193 |
| 45–54 | 11.4% | 1022 | 27.1% (24.5% to 29.7%) | 253 | 9.4% (9.0% to 9.7%) | 704 |
| 55–64 | 7.2% | 539 | 9.9% (7.8% to 12.0%) | 72 | 6.7% (6.4% to 7.0%) | 432 |
| 65+ | 9.8% | 747 | 4.7% (3.3% to 6.2%) | 38 | 10.0% (9.7% to 10.3%) | 652 |
| **Site** | | | | | | |
| Bonda Mission (subsistence farming area) | 15.9% (15.2% to 16.7%) | 1497 | 14.4% (12.0% to 16.8%) | 129 | 16.7% (15.8% to 17.5%) | 1343 |
| Eastern Highlands (tea estate) | 10.7% (10.0% to 11.3%) | 1019 | 13.1% (10.9% to 15.3%) | 117 | 10.8% (10.1% to 11.5%) | 888 |
| Selbourne (forestry estate) | 14.4% (13.7% to 15.1%) | 1377 | 15.6% (13.2% to 18.0%) | 145 | 14.8% (14.0% to 15.6%) | 1210 |
| Nyazura (town) | 14.8% (14.1% to 15.6%) | 1444 | 19.8% (17.1% to 22.4%) | 184 | 14.4% (13.7% to 15.3%) | 1211 |
| Nyanga (town) | 11.4% (10.8% to 12.0%) | 1157 | 6.8% (5.2% to 8.4%) | 69 | 10.0% (9.3% to 10.6%) | 866 |
| Watsomba (roadside settlement) | 17.4% (16.7% to 18.2%) | 1694 | 16.1% (13.8% to 18.5%) | 154 | 18.1% (17.2% to 18.9%) | 1496 |
| Sakubva (urban) | 7.8% (7.2% to 8.3%) | 789 | 8.6% (6.7% to 10.4%) | 82 | 7.5% (6.9% to 8.1%) | 654 |
| Hobhouse (urban) | 7.5% (7.0% to 8.1%) | 803 | 5.6% (4.2% to 7.1%) | 56 | 7.8% (7.2% to 8.3%) | 710 |
| **Employment** | | | | | | |
| Unemployed | 43.8 (42.9% to 44.7%) | 4322 | 47.6% (44.3% to 50.8%) | 472 | 43.4% (42.4% to 44.4%) | 4722 |
| Employed or in education | 56.2% (55.3% to 57.1%) | 5458 | 52.4% (49.2% to 55.7%) | 464 | 56.6% (55.6% to 57.6%) | 3656 |
| **Relationship status** | | | | | | |
| Married or in a long-term relationship | 59.5% (58.7% to 60.3%) | 5389 | 60.4% (57.3% to 63.6%) | 536 | 59.4% (58.5% to 60.3%) | 4594 |
| Never been in a long-term relationship | 25.9% (25.3% to 26.5%) | 3033 | 10.2% (8.4% to 12.1%) | 111 | 28.3% (27.7% to 29.0%) | 2821 |
| Separated or divorced | 6.8% (6.3% to 7.4%) | 663 | 13.2% (11.0% to 15.3%) | 131 | 5.8% (5.3% to 6.4%) | 483 |
| Widowed | 7.8% (7.3% to 8.2%) | 695 | 16.2% (13.8% to 18.5%) | 158 | 6.4% (5.9% to 6.9%) | 480 |
| **Education** | | | | | | |
| None or primary | 22.9% (22.2% to 23.7%) | 2023 | 27.1% (24.2% to 30.1%) | 244 | 22.4% (21.6% to 23.2%) | 1685 |
| Secondary or higher | 77.1% (76.3% to 77.8%) | 7595 | 72.9% (69.9% to 75.8%) | 663 | 77.6% (76.8% to 78.4%) | 6566 |
| Missing | – | 162 | – | 29 | – | 127 |

**Table 1** Continued

| | Total population (N=9780)* | | People living with HIV (N=936)* | | HIV-negative people (N=8378)* | |
|---|---|---|---|---|---|---|
| | Weighted proportion, % (95% CI) | N* | Weighted proportion, % (95% CI) | N* | Weighted proportion, % (95% CI) | N* |
| Asset-based wealth | | | | | | |
| 1 (poorest) | 19.9% (19.1% to 20.7%) | 1908 | 22.3% (19.6% to 25.1%) | 210 | 20.4% (19.5% to 21.3%) | 1667 |
| 2 | 19.7% (18.8% to 20.5%) | 1866 | 20.8% (18.1% to 23.5%) | 192 | 19.8% (18.9% to 20.7%) | 1609 |
| 3 | 20.7% (19.8% to 21.5%) | 1977 | 22.3% (19.6% to 25.1%) | 207 | 20.9% (20.0% to 21.8%) | 1710 |
| 4 | 19.9% (19.0% to 20.7%) | 1963 | 20.7% (18.0% to 23.4%) | 191 | 19.6% (18.7% to 20.4%) | 1662 |
| 5 (richest) | 19.9% (19.1% to 20.7%) | 1980 | 13.8% (11.6% to 16.1%) | 131 | 19.4% (18.5% to 20.3%) | 1653 |
| Missing | – | 86 | – | 5 | – | 77 |

*Values are unweighted. In addition, HIV status was missing for some participants, so the unweighted Ns for people living with HIV and HIV-negative people do not sum to the total population unweighted N.
†CIs are not shown for variables used in weighting.
ART, antiretroviral therapy.

high (43.8%, 95% CI: 42.9% to 44.7%), yet most participants had secondary education or higher (75.7%, 74.9% to 76.4%).

### Association between HIV and self-reported hypertension

In this population, the weighted prevalence of self-reported hypertension was 13.6% (12.9% to 14.2%) and the weighted prevalence of HIV was 11.1% (10.4% to 11.7%). The prevalence of self-reported hypertension in PLHIV (14.1%, 11.9% to 16.3%) was similar to the prevalence of self-reported hypertension in HIV-negative people in crude univariable analyses (13.3%, 12.6% to 14.0%, F=0.4, p=0.503). The lack of evidence for a difference in univariable analyses remained when analysing the prevalence of self-reported hypertension within age and sex groups for most groups (figure 2, online supplemental table 1). However, there was statistical support for lower prevalence of self-reported hypertension in PLHIV than in HIV-negative people among women aged 45–54 (PLHIV: 20.2%, 13.8% to 26.7%; HIV-negative: 30.1%, 25.7% to 34.5%; F=5.4, p=0.021) and men aged over 65 (PLHIV: 6.7%, 0.0% to 19.3%; HIV-negative: 31.2%, 25.3% to 37.2%, F=4.1, p=0.043) in univariable analyses.

In the initial multivariable logistic regression analysis adjusted for age and sex, no evidence was found for an association between HIV and self-reported hypertension (OR=0.87, 95% CI 0.70 to 1.07, p=0.191). Introducing further confounders (employment, relationship status, wealth and site) to generate a fully adjusted multivariable model did not alter this finding (0.88, 0.70 to 1.10, p=0.261; table 2).

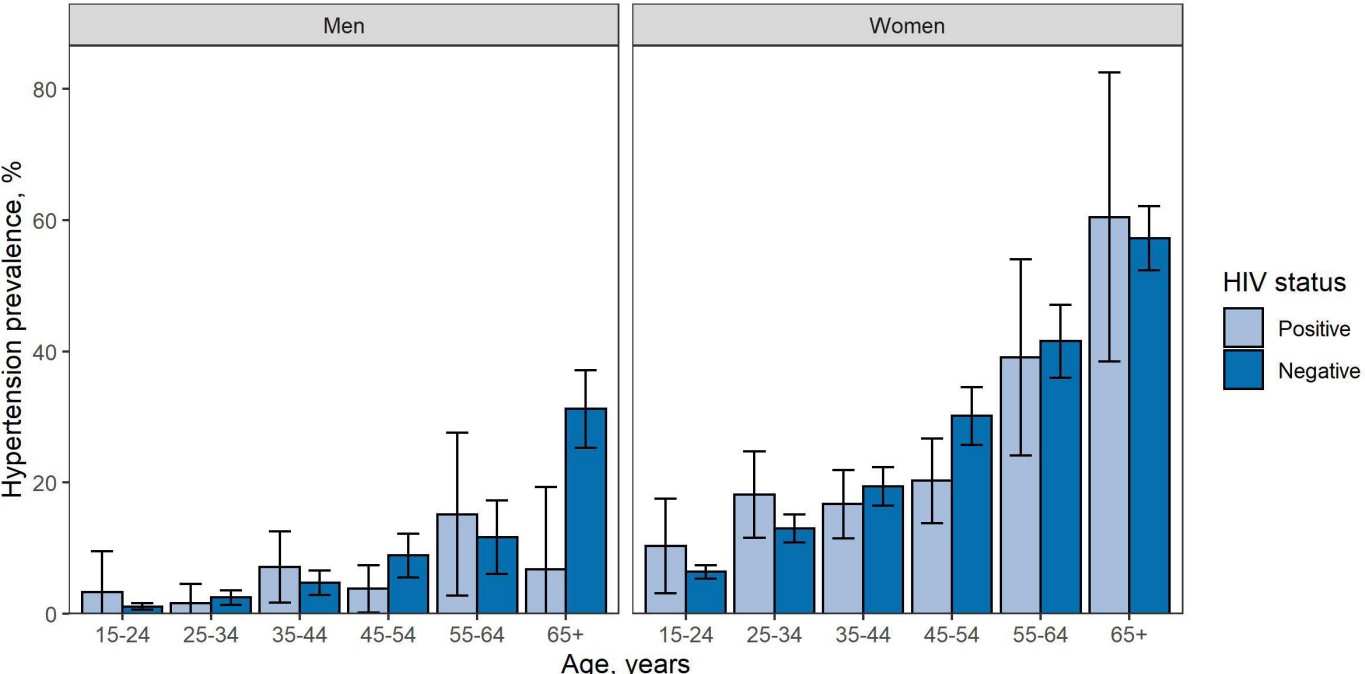

**Figure 2** Prevalence of reported hypertension in men and women by age and HIV status. Error bars: 95% CIs.

**Table 2** HIV as a determinant of hypertension

| | Weighted hypertension prevalence, % (95% CI) | OR adjusted for only age and sex (95% CI) | P value | Fully adjusted OR (95% CI) | P value |
|---|---|---|---|---|---|
| HIV status | | | 0.191 | | 0.261 |
| Negative | 13.3% (12.6% to 14.0%) | 1 | | 1 | |
| Positive | 14.1% (11.9% to 16.3%) | 0.87 (0.70 to 1.07) | | 0.88 (0.70 to 1.10) | |
| Sex | | | <0.001 | | <0.001 |
| Male | 6.7% (5.9% to 7.6%) | 1 | | 1 | |
| Female | 19.8% (18.8% to 20.8%) | 4.37 (3.71 to 5.15) | | 4.48 (3.70 to 5.41) | |
| Age, years | | | <0.001 | | <0.001 |
| 15–24 | 3.9% (3.3% to 4.5%) | 1 | | 1 | |
| 25–34 | 8.2% (7.1% to 9.3%) | 2.24 (1.79 to 2.80) | | 1.97 (1.50 to 2.59) | |
| 35–44 | 12.2% (10.7% to 13.8%) | 3.57 (2.87 to 4.34) | | 3.24 (2.45 to 4.27) | |
| 45–54 | 17.6% (15.4% to 19.8%) | 5.79 (4.63 to 7.26) | | 5.70 (4.28 to 7.59) | |
| 55–64 | 27.6% (24.0% to 31.2%) | 10.93 (8.49 to 14.07) | | 10.90 (7.92 to 15.01) | |
| 65 and over | 43.7% (40.1% to 47.2%) | 24.32 (19.37 to 30.53) | | 26.77 (19.69 to 36.38) | |
| Site | | | <0.001 | | <0.001 |
| Bonda Mission (subsistence farming area) | 14.1% (12.3% to 15.9%) | 1 | | 1 | |
| Eastern Highlands (tea estate) | 6.1% (4.6% to 7.6%) | 0.92 (0.66 to 1.29) | | 0.83 (0.59 to 1.18) | |
| Selbourne (forestry estate) | 9.4% (7.8% to 11.0%) | 1.04 (0.79 to 1.39) | | 1.16 (0.87 to 1.54) | |
| Nyazura (town) | 13.3% (11.4% to 15.1%) | 1.40 (1.08 to 1.82) | | 1.32 (1.01 to 1.72) | |
| Nyanga (town) | 9.3% (7.5% to 11.1%) | 1.09 (0.82 to 1.46) | | 0.80 (0.57 to 1.12) | |
| Watsomba (roadside settlement) | 16.9% (15.0% to 18.7%) | 1.43 (1.13 to 1.80) | | 1.48 (1.16 to 1.88) | |
| Sakubva (urban) | 28.2% (24.9% to 31.5%) | 5.28 (4.03 to 6.91) | | 4.36 (3.21 to 5.90) | |
| Hobhouse (urban) | 15.5% (12.8% to 18.1%) | 2.73 (2.03 to 3.66) | | 1.93 (1.38 to 2.70) | |
| Employment | | | 0.001 | | 0.006 |
| Unemployed | 19.5% (18.3% to 20.6%) | 1 | | 1 | |
| Employed or in education | 9.0% (8.2% to 9.8%) | 0.79 (0.68 to 0.91) | | 0.80 (0.69 to 0.94) | |
| Relationship status | | | <0.001 | | <0.001 |
| Married or in a long-term relationship | 14.6% (13.6% to 15.5%) | 1 | | 1 | |
| Never been in a long-term relationship | 3.9% (3.2% to 4.7%) | 0.50 (0.38 to 0.66) | | 0.54 (0.41 to 0.73) | |
| Separated or divorced | 15.8% (13.1% to 18.6%) | 0.92 (0.72 to 1.18) | | 1.03 (0.79 to 1.33) | |
| Widowed | 35.9% (32.3% to 39.4%) | 0.67 (0.54 to 0.83) | | 0.74 (0.58 to 0.95) | |
| Education | | | <0.001 | | – |
| None or primary | 24.1% (22.3% to 26.0%) | 1 | | – | |
| Secondary or higher | 10.1% (9.5% to 10.9%) | 1.52 (1.24 to 1.86) | | – | |
| Asset-based wealth | | | <0.001 | | <0.001 |
| 1 (poorest) | 11.8% (10.3% to 13.3%) | 1 | | 1 | |
| 2 | 11.4% (9.9% to 12.9%) | 1.09 (0.86 to 1.39) | | 1.15 (0.89 to 1.47) | |
| 3 | 13.0% (11.5% to 14.6%) | 1.38 (1.10 to 1.74) | | 1.40 (1.10 to 1.79) | |
| 4 | 15.8% (14.1% to 17.4%) | 2.11 (1.69 to 2.65) | | 1.52 (1.17 to 1.98) | |
| 5 (richest) | 16.0% (14.3% to 17.7%) | 2.31 (1.84 to 2.90) | | 2.11 (1.61 to 2.76) | |

## Association between ART-exposure and self-reported hypertension

Over half of PLHIV were ART-exposed (65.8%, 62.8% to 69.0%, table 1). The prevalence of self-reported hypertension was similar among ART-exposed PLHIV (14.8%, 12.0% to 17.7%) and ART-naïve PLHIV in crude univariable comparisons (12.8%, 9.1% to 16.4%, F=0.7, p=0.388) and the lack of a difference remained in univariable analyses of the prevalence of self-reported hypertension split by age and sex (online supplemental figure 1).

In multivariable logistic regression analysis, there was no evidence of a relationship between ART-exposure and self-reported hypertension after adjustment for age and sex (OR=0.87, 95% CI 0.56 to 1.33, p=0.519) or after further adjustment for site (0.83, 0.53 to 1.30, p=0.411; table 3).

**Table 3** Antiretroviral therapy as a determinant of hypertension in people living with HIV

| | Weighted hypertension prevalence, % (95% CI) | OR adjusted for only age and sex (95% CI) | P value | Fully adjusted OR (95% CI) | P value |
|---|---|---|---|---|---|
| ART exposure | | | 0.519 | | 0.411 |
| ART-naïve | 12.8% (9.1% to 16.4%) | 1 | | 1 | |
| ART-exposed | 14.8% (12.0% to 17.7%) | 0.87 (0.56 to 1.33) | | 0.83 (0.53 to 1.30) | |
| Sex | | | <0.001 | | <0.001 |
| Male | 6.0% (3.3% to 8.8%) | 1 | | 1 | |
| Female | 20.3% (17.1% to 23.6%) | 5.11 (2.95 to 8.84) | | 5.73 (3.23 to 10.17) | |
| Age, years | | | <0.001 | | <0.001 |
| 15–24 | 7.9% (2.6% to 13.1%) | 1 | | 1 | |
| 25–34 | 12.3% (7.8% to 16.8%) | 1.69 (0.72 to 3.94) | | 2.03 (0.86 to 4.80) | |
| 35–44 | 13.1% (9.2% to 17.0%) | 1.87 (0.83 to 4.23) | | 2.58 (1.12 to 5.93) | |
| 45–54 | 12.0% (8.2% to 15.8%) | 2.01 (0.88 to 4.58) | | 2.80 (1.19 to 6.59) | |
| 55–64 | 24.7% (14.7% to 34.6%) | 6.19 (2.43 to 15.75) | | 9.54 (3.43 to 26.49) | |
| 65 and over | 27.8% (13.8% to 41.9%) | 7.53 (2.71 to 20.94) | | 12.51 (4.16 to 37.67) | |
| Site | | | <0.001 | | <0.001 |
| Bonda Mission (subsistence farming area) | 13.3% (7.8% to 18.9%) | 1 | | 1 | |
| Eastern Highlands (tea estate) | 6.1% (2.0% to 10.2%) | 0.64 (0.26 to 1.57) | | 0.64 (0.26 to 1.56) | |
| Selbourne (forestry estate) | 8.3% (3.9% to 12.6%) | 0.82 (0.37 to 1.82) | | 0.81 (0.37 to 1.80) | |
| Nyazura (town) | 19.6% (13.5% to 25.8%) | 2.13 (1.10 to 4.14) | | 2.15 (1.10 to 4.17) | |
| Nyanga (town) | 6.1% (0.1% to 12.0%) | 0.44 (0.13 to 1.45) | | 0.43 (0.13 to 1.44) | |
| Watsomba (roadside settlement) | 10.6% (5.8% to 15.3%) | 0.84 (0.42 to 1.67) | | 0.82 (0.41 to 1.64) | |
| Sakubva (urban) | 30.9% (20.9% to 40.8%) | 4.58 (2.21 to 9.51) | | 4.53 (2.19 to 9.41) | |
| Hobhouse (urban) | 25.8% (14.4% to 37.3%) | 3.29 (1.43 to 7.54) | | 3.27 (1.43 to 7.50) | |
| Employment | | | 0.352 | | – |
| Unemployed | 16.0% (12.7% to 19.3%) | 1 | | – | |
| Employed or in education | 12.3% (9.3% to 15.4%) | 1.20 (0.81 to 1.78) | | – | |
| Relationship status | | | 0.671 | | – |
| Married or in a long-term relationship | 12.4% (9.6% to 15.2%) | 1 | | – | |
| Never been in a long-term relationship | 11.1% (4.2% to 18.0%) | 0.81 (0.33 to 2.03) | | – | |
| Separated or divorced | 17.2% (10.9% to 23.4%) | 1.17 (0.70 to 1.95) | | – | |
| Widowed | 19.8% (13.6% to 25.9%) | 0.81 (0.50 to 1.31) | | – | |
| Education | | | 0.562 | | – |
| None or primary | 17.6% (12.8% to 22.4%) | 1 | | – | |
| Secondary or higher | 12.6% (10.0% to 15.1%) | 1.15 (0.72 to 1.81) | | – | |
| Asset-based wealth | | | 0.003 | | – |
| 1 (poorest) | 8.8% (5.1% to 12.5%) | 1 | | – | |
| 2 | 12.0% (7.5% to 16.5%) | 1.67 (0.88 to 3.16) | | – | |
| 3 | 12.6% (8.0% to 17.2%) | 1.64 (0.86 to 3.15) | | – | |
| 4 | 18.0% (12.7% to 23.3%) | 2.75 (1.49 to 5.09) | | – | |
| 5 (richest) | 22.9% (15.3% to 30.5%) | 3.15 (1.64 to 6.07) | | – | |

ART, antiretroviral therapy.
ART, antiretroviral therapy.

Results for sensitivity analyses focused on current ART usage and adherent current ART usage were similar to those for ART exposure (online supplemental tables 2–4). Fully adjusted multivariable logistic regression models showed no evidence of a relationship between current ART usage and hypertension (0.80, 0.51 to 1.25, p=0.312; online supplemental table 3), or adherent current ART usage and hypertension (0.70, 0.46 to 1.08, p=0.107; online supplemental table 4).

## DISCUSSION

This study aimed to examine whether HIV status and ART exposure were associated with self-reported hypertension

in Manicaland, East Zimbabwe. In this setting, approximately one in seven PLHIV self-reported hypertension. There was no evidence of a difference in the prevalence of self-reported hypertension by HIV status. Furthermore, no evidence was found for an association between ART-exposure and self-reported hypertension among PLHIV.

The results of this study corroborate previous estimates of the prevalence of hypertension in Zimbabwean PLHIV. The prevalence of self-reported hypertension in PLHIV in Manicaland was 14.1% (11.9% to 16.3%), which is similar to earlier estimates of 8.9%, 18.3% and 19.8% from elsewhere in the country.[14–16] The alignment of this study's findings with previous estimates indicates that the results may be a realistic representation of the prevalence of hypertension. However, this means that there is already an important burden of hypertension in PLHIV in Zimbabwe, which is likely to grow as the population grows older and unhealthy lifestyles become more widespread.[30 31] The use of newer integrase inhibitors may also increase weight gain among PLHIV and raise the prevalence of hypertension further.[32 33] Consequently, more prevention and treatment services may be required to avoid worsening outcomes.

Knowledge of the burden of hypertension experienced by PLHIV is key to providing prevention and treatment services as efficiently as possible, particularly given the scarce resources available.[31 34] This study found a similar prevalence of self-reported hypertension among PLHIV and HIV-negative people. These findings differ from those from a systematic review, which pooled prevalence estimates gathered across Africa and indicated that PLHIV might experience a lower prevalence of hypertension.[6] The difference may be explained in part by regional variation in the relationship between HIV and hypertension within Africa and in part due to the different approaches to detection of hypertension that were used. The systematic review focused on studies reporting clinical diagnoses of hypertension, excluding data on self-reported hypertension as is presented here.[6] Further, a study of data from the Agincourt Health and Demographic Surveillance System in South Africa, which neighbours Zimbabwe, also reported that the prevalence of hypertension among PLHIV was comparable to that of HIV-negative people.[35] The South African study and the global systematic review suggest that regional differences may exist, which should be fully accounted for when developing guidelines for scale-up of hypertension services.[6 35] Notably, it will be important to identify and manage other risk factors that are associated with hypertension in Zimbabwe.

Our findings further indicated that the prevalence of self-reported hypertension was similar among ART-exposed and ART-naïve PLHIV. Several large studies, including the Data Collection on Adverse Events of Anti-HIV Drugs (D:A:D) study, have obtained similar results, but a previous study in Zimbabwe reported a higher prevalence of hypertension among PLHIV on ART.[14 36] The conflicting results from the two Zimbabwean studies may be due to differences in how ART-exposure

was measured. While this study focused on whether PLHIV had ever taken ART, the previous Zimbabwean study examined the effect of taking ART for more than 5 years.[14] Detailed data on ART interruption and discontinuation over time are not available from the Manicaland Study. However, additional research examining how patterns of ART-exposure over time relate to hypertension risk might clarify the source of these differences and give further insight into how hypertension services should be delivered.

### Strengths and limitations

To our knowledge, this is the first study to provide estimates of the prevalence of hypertension among comparable populations of PLHIV and HIV-negative people in a non-clinical setting in Zimbabwe. This study also used robust methodology to explore links between HIV status, ART-exposure and self-reported hypertension, building the evidence base on links between HIV and non-communicable diseases. In addition, the use of data from a large population survey should have increased the generalisability of the findings to other similar settings in Zimbabwe. Finally, all analyses were underpinned by individual-level data that included a diverse range of men and women and could be weighted to local census data, which enhanced the representativeness of results.

However, this work also has some limitations. These fall across five broad areas: the definition of hypertension, unmeasured risk factors for hypertension, selection biases towards the healthy, the use of a large number of significance tests and the cross-sectional nature of the analysis. First, funding limitations led to a reliance on self-report of previous diagnoses by a doctor or nurse as a proxy to detect disease, which may have resulted in underestimation of the underlying hypertension burden.[37] Limited access to screening and diagnosis will likely have meant that some hypertensive people in our study population were at the early stages of the hypertension care cascade and unaware of their hypertension, and so were excluded from our estimates.[38–41] Notably, a 2015 meta-analysis of four studies indicated that the prevalence of hypertension in the general population in Zimbabwe may be 30%, which is higher than our estimates of the prevalence of self-reported hypertension among PLHIV and HIV-negative people. The difference between the two reports suggests that a substantial proportion of the participants in this study could have been unaware that they had hypertension and so were not included in our estimates.[42] This would reduce the generalisability of our findings for settings in which hypertension screening and diagnosis is more readily available. In addition, HIV-negative people may have had less contact with the healthcare system for hypertension diagnosis than PLHIV, which could have artificially reduced self-reporting of hypertension in this group.[43] Yet the prevalence of hypertension was comparable among PLHIV who knew their status (14.7%, 11.9% to 17.5%) and PLHIV who did not (15.1%, 10.6% to 19.5%), suggesting that contact with the healthcare

system among diagnosed PLHIV did not increase self-reporting of hypertension. In future survey rounds, quantification of the prevalence of hypertension by measuring participants' blood pressure during the individual interview, and asking questions to detect medications for hypertension, would provide more reliable estimates of this burden.[44] The results of this analysis could then be updated accordingly.

As well as limitations linked to the definition of hypertension, data were not available on several risk factors for hypertension, such as body mass index (BMI), diet and physical activity, which precluded their use in the analysis.[45 46] Future studies should measure the height and weight of participants and calculate their BMI, and gather data on diet and physical activity levels among participants so that analyses can be adjusted for these risk factors. Moreover, people with hypertension, or conditions caused by hypertension, were more likely to be seriously ill and so unable to complete the individual interview, or in locations, such as hospitals, which were not reached by the field teams. In addition, the analysis involved a large number of significance tests, without correcting for multiple comparisons, which can increase the probability of a false positive finding due to chance. Finally, assessment of the association between HIV and ART and self-reported hypertension was constrained by the cross-sectional nature of the analysis. The development of a cohort that consisted of matched PLHIV and HIV-negative controls in Manicaland, which could be followed over time, would allow more certainty in the drawing of conclusions about the relationship.[14]

## Conclusion

Approximately one in seven PLHIV self-reported hypertension in Manicaland, Zimbabwe. However, HIV status and ART exposure were not associated with self-reported hypertension. Zimbabwe is reorienting its health system towards control of hypertension and other noncommunicable diseases; this study highlights how carefully designed, sufficiently resourced prevention and care interventions for PLHIV and HIV-negative people will be required to make this a success.

**Acknowledgements** The authors would like to express our gratitude to everyone who has been involved with the Manicaland Study over the past 20 years, especially the participants and research staff. Without them, this study would not exist.

**Contributors** MS, SG and KD conceived the analysis and developed the methods. SG, CN, RM, PM and TM led the data collection and TD and LM cleaned, extracted and prepared the data. KD performed the analysis and generated figures and estimates. KD led the writing of the manuscript with MS and SG, and all authors commented on and contributed to the finalisation of the manuscript. KD is responsible for the overall content as the guarantor.

**Funding** This work was supported by the Wellcome Trust [220098/Z/20/Z to KD]; the Bill and Melinda Gates Foundation [OPP1161471 to SG]; the Civilian Research and Development Foundation Global (prime is President's Emergency Plan for AIDS Relief) [OISE-9531011 to MS]; the National Institutes of Health [5R01MH114562-02 to SG and CN; 1R21AG053093-01 to MS] and the Medical Research Council Centre for Global Infectious Disease Analysis [MR/R015600/1 to KD, LM, CN, MS

and SG], which is jointly funded by the UK Medical Research Council and the UK Foreign, Commonwealth & Development Office (FCDO), under the MRC/FCDO Concordat agreement and is also part of the EDCTP2 programme supported by the European Union. The funders had no role in study design, data collection and analysis, decision to publish, or preparation of the manuscript. For the purpose of open access, the author has applied a CC BY public copyright licence to any Author Accepted Manuscript version arising from this submission.

**Competing interests** SG declares holding shares in AstraZeneca and GlaxoSmithKline. All other authors declare that no competing interests exist.

**Patient and public involvement** Patients and/or the public were involved in the design, or conduct, or reporting, or dissemination plans of this research. Refer to the Methods section for further details.

**Patient consent for publication** Not applicable.

**Ethics approval** The Manicaland Study was conducted with the understanding and informed consent of participants. Ethical approval was given by Imperial College Research Ethics Committee (17IC4160), the Biomedical Research and Training Institute Institutional Review Board (AP140/2017) and the Medical Research Council of Zimbabwe (MRCZ/A/2243).

**Provenance and peer review** Not commissioned; externally peer reviewed.

**Data availability statement** Data are available upon reasonable request. The data are deidentified participant data and can be obtained by sending a data request form (available from the Manicaland Study website: http://www.manicalandhivproject.org/data.html) to s.gregson@imperial.ac.uk.

**ORCID iD**
Katherine Davis http://orcid.org/0000-0002-3291-649X

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
