## [Reviewer comments · BMJ Open]

ARTICLE DETAILS

TITLE (PROVISIONAL)	Associations between HIV status and self-reported hypertension in a high HIV prevalence sub-Saharan African population: A cross-sectional study
AUTHORS	Davis, Katherine; Moorhouse, Louisa; Maswera, Rufurwokuda; Mandizvidza, Phyllis; DADIRAI, Tawanda; Museka, Tafadzwa; Nyamukapa, Constance; Smit, Mikaela; Gregson, Simon

VERSION 1 – REVIEW

REVIEWER	Shamu, Tinei Newlands Clinic
REVIEW RETURNED	21-Sep-2022

GENERAL COMMENTS	Associations between HIV status and self-reported hypertension in a high HIV prevalence sub-Saharan African population: A cross-sectional study This is a cross sectional study conducted in the Eastern Highlands of Zimbabwe in a population predominantly from farming areas but also includes urban locations. The authors sought to investigate associations between HIV status as tested during the study and self-reported hypertension (HTN) i.e., tested and verified by a doctor or nurse prior to the survey. Prevalence of self-reported HTN was found to be similar among PLWH and HIV negative participants. The authors note that the observed HTN prevalence may be underestimated as it is a function of access to healthcare services. In addition, the authors also investigated the association between ART exposure among PLWH and HTN. Again, there was no evidence of an association. The manuscript is well written and reads clearly. All steps are clearly described, and the appropriate reporting guideline (STROBE) was followed with the checklist appended. Statistical analysis The statistical analysis implemented in the determination of associations is clearly described. However, the authors also interpret the adjusted odds ratios of the covariates (confounders), and here it is unclear whether the same model designed for the measurement of association between HIV status and HTN is the same model interpreted for the confounders. This may introduce Table 2 fallacy as such a model was not primarily built to assess age or cis gender for example. A clarification of whether separate models were built to this end or the same one was used (and the justification why this would have been done) may be helpful in the interpretation of the aORs. Exposure and outcome The choice of using only self-reported HTN must have been convenient, but in my view is a serious limitation within the context of the study population. It might be helpful to state why the choice
---

	was made to only use self-reported HTN and not screen. While some studies were cited showing prevalence of HTN among PLWH, an important meta-analysis of HTN in Zimbabwe (doi: 10.13105/wjma.v3.i1.54 Mutowo MP, Mangwiro JC, Lorgelly P, Owen A, Renzaho AM. Hypertension in Zimbabwe: A meta-analysis to quantify its burden and policy implications. World J Meta-Anal 2015; 3(1): 54-60 [DOI: 10.13105/wjma.v3.i1.54]) was not cited that showed a pooled prevalence of 30% in the general population . The difference between the observed prevalence and the pooled estimate from this meta-analysis warrants some discussion.
--	---

REVIEWER	Murray, Melanie British Columbia Women's Hospital and Health Centre Women's Health Research Institute
REVIEW RETURNED	24-Oct-2022

GENERAL COMMENTS	With the widespread availability of cART to treat HIV infection, comorbid illnesses are becoming increasingly important to diagnose and treat. Hypertension is a common comorbid illness in the general population, though data are conflicting re: whether this comorbidity is more common among people living with HIV. The authors herein seek to examine hypertension among a large sample of persons living in various settings in Zimbabwe. Comments: General: 1) The authors refer to analyses as having been done by “gender – male, female” but the questionnaire asks for the respondents’ “sex – i.e. male or female”. Please amend the paper to refer to sex throughout (sex, female, male) unless there is another question that was used. If there was another question used, please point this out. Introduction: 2) References 4 and 5 are not studies showing higher rates of hypertension. Please cite the actual evidence. 3) Page 5, paragraph 2, last sentence: Please change “...has not been confirmed in this setting” to “has not been replicated in this setting”. 4) Some argument as to the proposed reasoning for there to be a difference in hypertension rates between positive and negative individuals would be beneficial. Methods: 5) Analysis would have benefitted from assessing current ART use as well as “ever” use. It is not clear to me what mechanism would be responsible for hypertension in someone who has taken it at one time (perhaps even years ago, perhaps only surrounding labour and delivery for example) but who is not currently on ART. Results: 6) Page 15, line 38: Please change “In logistic regression analysis, there was not strong evidence of a relationship....” To “In logistic regression analysis, there was no evidence of a relationship....” 7) Many comparisons are quoted in this paper, and there is no discussion of correcting for multiple comparisons. Further, it is difficult to differentiate between univariate and multivariate results as they are stated in the text – further adding to the sense that many comparisons were made.
--

	Discussion: 8) Page 18, line 9-13: “A variety of physiological.....vital to further understanding”. This would be well-suited to the introduction – in fact – some argument as to the proposed reasoning for there to be a difference in hypertension rates between positive and negative individuals would be beneficial. 9) Page 18, discussion in first paragraph: Another possibility is that those living with HIV may have been more likely to have been told they had hypertension when they went to their physician/clinic to obtain medication or other care for their HIV. Those not living with HIV may have encountered health care providers less frequently, and had their blood pressure checked less frequently resulting in an under-reporting of hypertension in this group compared to the group living with HIV. This point needs to be discussed here. 10) Page 18, discussion in second paragraph: This reviewer would argue that newer Integrase inhibitors which have been associated with weight gain could also play a role in hypertension – as increased weight is associated with increased rate of hypertension in the general population. Please amend accordingly. 11) Page 19, line 18: NCD – This term is defined at the very beginning of the article, then not used until the end. Please remove and write out in full. 12) Page 19, line 48-52: Measurement of blood pressure, and questions to detect medications for hypertension would also be important. 13) Page 20, lines 4-8: BMI is also an important factor affecting hypertension. Height and weight should also be recorded in future studies and analyses adjusted for this. 14) Please include discussion of additional limitations to the generalizability of this study:  a. The Study population is more educated and of higher socioeconomic status than average in the country. The study included only 3% of those from the lowest socioeconomic status and is therefore not generalizable to that group. b. Women had higher odds of self-reported hypertension – could this be related to frequency of visits to a physician rather than a true increase in prevalence? c. Wealthiest – higher odds of reporting hypertension than poor – and city vs. subsistence farming area – is this related to access to care (i.e. such that those who are more poor and rural are less likely to access a care provider to ever be told they have hypertension)?
--	---

VERSION 1 – AUTHOR RESPONSE

Reviewer: 1

Dr. Tinei Shamu, Newlands Clinic

This is a cross sectional study conducted in the Eastern Highlands of Zimbabwe in a population predominantly from farming areas but also includes urban locations. The authors sought to investigate associations between HIV status as tested during the study and self-reported hypertension (HTN) i.e., tested and verified by a doctor or nurse prior to the survey. Prevalence of self-reported HTN was found to be similar among PLWH and HIV negative participants. The authors note that the observed HTN prevalence may be underestimated as it is a function of access to healthcare services. In addition, the

authors also investigated the association between ART exposure among PLWH and HTN. Again, there was no evidence of an association.

The manuscript is well written and reads clearly. All steps are clearly described, and the appropriate reporting guideline (STROBE) was followed with the checklist appended.

Response: We thank the reviewer for highlighting the strengths of our manuscript. We have responded to each of the comments below.

Comments

Statistical analysis

The statistical analysis implemented in the determination of associations is clearly described. However, the authors also interpret the adjusted odds ratios of the covariates (confounders), and here it is unclear whether the same model designed for the measurement of association between HIV status and HTN is the same model interpreted for the confounders. This may introduce Table 2 fallacy as such a model was not primarily built to assess age or cis gender for example. A clarification of whether separate models were built to this end or the same one was used (and the justification why this would have been done) may be helpful in the interpretation of the aORs.

Response: The same model was used for assessment of the association between HIV status and hypertension, and to inform the interpretation of the confounders. We agree that this may introduce a Table 2 fallacy, because the model was not specifically designed for assessing the association between the confounders and hypertension. As a result, we have removed the interpretation of the adjusted odds ratios for the confounders from the text, to keep the manuscript focused on the primary effect.

Exposure and outcome

The choice of using only self-reported HTN must have been convenient, but in my view is a serious limitation within the context of the study population. It might be helpful to state why the choice was made to only use self-reported HTN and not screen.

Response: We appreciate that this is a limitation of the current study. The main objectives of the 2018-2019 round of the Manicaland Study were related to HIV risk perception, rather than non-communicable diseases. This meant that funding for detecting hypertension was limited and screening was not possible. Nonetheless, we believe our approach to assessing hypertension has value. This is in part because using a simple method based on a question about previous diagnosis allowed us to rapidly assess self-reported hypertension among a substantial number of people in Manicaland. In addition, there is very limited data on hypertension in Manicaland, so self-reported data can still provide useful insights into the burden of hypertension.

In line with the reviewer's suggestion, we have clarified the reason for using self-reported hypertension, rather than screening, in the text.

Methods

"Funding limitations meant that blood pressure was not measured during the household census or individual interview."

Discussion

"Firstly, funding limitations led to a reliance on self-report of previous diagnoses by a doctor or nurse as a proxy to detect disease, which may have resulted in underestimation of the underlying hypertension burden."

While some studies were cited showing prevalence of HTN among PLWH, an important meta-analysis of HTN in Zimbabwe (doi: 10.13105/wjma.v3.i1.54 Mutowo MP, Mangwiro JC, Lorgelly P, Owen A, Renzaho AM. Hypertension in Zimbabwe: A meta-analysis to quantify its burden and policy implications. World J Meta-Anal 2015; 3(1): 54-60 [DOI: 10.13105/wjma.v3.i1.54]) was not cited that showed a pooled prevalence of 30% in the general population. The difference between the observed prevalence and the pooled estimate from this meta-analysis warrants some discussion.

Response: Thank you for sharing this reference. We have added a discussion of the difference between the observed prevalence figures and the pooled estimate from the meta-analysis analysis to the Strengths and limitations section of the paper.

Discussion – Strengths and limitations

“Firstly, funding limitations led to a reliance on self-report of previous diagnoses by a doctor or nurse as a proxy to detect disease, which may have resulted in underestimation of the underlying hypertension burden.[37] Limited access to screening and diagnosis will likely have meant that some hypertensive people in our study population were at the early stages of the hypertension care cascade and unaware of their hypertension, and so were excluded from our estimates.[38-41] Notably, a 2015 meta-analysis of four studies indicated that the prevalence of hypertension in the general population in Zimbabwe may be 30%, which is higher than our estimates of the prevalence of self-reported hypertension among PLHIV and HIV-negative people. The difference between the two reports suggests that a substantial proportion of the participants in this study could have been unaware that they had hypertension and so were not included in our estimates [42].”

Reviewer: 2

Dr. Melanie Murray, British Columbia Women's Hospital and Health Centre Women's Health Research Institute, The University of British Columbia Faculty of Medicine

With the widespread availability of cART to treat HIV infection, comorbid illnesses are becoming increasingly important to diagnose and treat. Hypertension is a common comorbid illness in the general population, though data are conflicting re: whether this comorbidity is more common among people living with HIV. The authors herein seek to examine hypertension among a large sample of persons living in various settings in Zimbabwe.

Response: We agree with the reviewer about the importance of diagnosing and treating comorbid illnesses among people living with HIV. We thank them for their comments on the manuscript, which we have responded to below.

Comments**General**

- 1 The authors refer to analyses as having been done by "gender – male, female" but the questionnaire asks for the respondents' "sex – i.e., male or female". Please amend the paper to refer to sex throughout (sex, female, male) unless there is another question that was used. If there was another question used, please point this out.

Response: We agree that this should be stated as sex, and we have amended the paper through-out.

Abstract

"After weighting of survey responses by age and sex using household census data, chi-squared tests and logistic regression were used to explore whether HIV status and ART-exposure were associated with self-reported hypertension."

Methods – Weighting

"First, census data were analysed to calculate the proportion of the population in 5-year age groups between 15 and 74 years, as well as a 75+ year-old age-group, by sex. Iterative proportional fitting, also known as raking, was then used to generate weights that matched the age and sex proportions in the interview sample to those in the household census. Participant data were weighted for all further analyses; this approach meant that 42 people with missing age or sex information were excluded."

Methods - Association between HIV and self-reported hypertension

"To compare the prevalence of self-reported hypertension between PLHIV and HIV-negative people, estimates for the prevalence of self-reported hypertension stratified by HIV status, age and sex were produced in the same manner."

"Crude comparisons of the prevalence of self-reported hypertension between PLHIV and HIV-negative people were made using Chi-squared tests, with Rao-Scott adjustment. These were performed both for the overall population and within age and sex strata."

"All models were complete case analyses. Initially, a model was built including HIV, age, and sex."

"Second, any potential confounders that were found to be associated with self-reported hypertension in models including age and sex, at a threshold p-value of 0.1, were added to the initial model."

Methods - Association between ART-exposure and self-reported hypertension

“Prevalence estimates for self-reported hypertension, stratified by ART-exposure, age, and sex, were then produced and a crude comparison between ART-exposed and ART-naïve PLHIV was made using Chi-squared tests, with Rao-Scott adjustment.”

Results

“After exclusion of people without age or sex data, there were 9,780 participants in the study population.”

Results - Association between HIV and self-reported hypertension

“The lack of evidence for a difference in univariable analyses remained when analysing the prevalence of self-reported hypertension within age and sex groups for most groups (Fig. 2, Supplementary table 1).”

“In the initial multivariable logistic regression analysis adjusted for age and sex, no evidence was found for an association between HIV and self-reported hypertension (odds ratio, OR=0.87, 95% CI=0.70-1.07, p=0.191).”

Results - Association between ART-exposure and self-reported hypertension

“The prevalence of self-reported hypertension was similar among ART-exposed PLHIV (14.8%, 12.0%-17.7%) and ART-naïve PLHIV in crude univariable comparisons (12.8%, 9.1%-16.4%, F=0.7, p=0.388) and the lack of a difference remained in univariable analyses of the prevalence of self-reported hypertension split by age and sex (Supplementary fig. 1).”

“In multivariable logistic regression analysis, there was no evidence of a relationship between ART-exposure and self-reported hypertension after adjustment for age and sex (OR=0.87, 95% CI=0.56-1.33, p=0.519) or after further adjustment for site (0.83, 0.53-1.30, p=0.411; Table 3).”

Tables, figures, and supplementary material have also been updated.

Introduction

- 2 References 4 and 5 are not studies showing higher rates of hypertension. Please cite the actual evidence.

Response: We have added three further references to better present the evidence on differing rates of hypertension.[1,2,3] Two of the new references report original research comparing the burden of hypertension among people living with HIV to HIV-negative controls. The final additional reference is a systematic review and meta-analysis of studies examining hypertension risk by HIV status.

1. van Zoest RA, Wit FW, Kooij KW, *et al.* Higher Prevalence of Hypertension in HIV-1-Infected Patients on Combination Antiretroviral Therapy Is Associated With Changes in Body Composition and Prior Stavudine Exposure. *Clin Infect Dis* 2016;**63**:205–13. doi:10.1093/cid/ciw285
2. Mayer KH, Loo S, Crawford PM, *et al.* Excess clinical comorbidity among HIV-infected patients accessing primary care in US community health centers. *Public Health Rep* 2018;**133**:109–18. doi:10.1177/0033354917748670
3. Davis K, Perez-Guzman P, Hoyer A, *et al.* Association between HIV infection and hypertension: a global systematic review and meta-analysis of cross-sectional studies. *BMC Med* 2021;**19**:105. doi:10.1186/s12916-021-01978-7

Introduction

“Globally, there is some evidence that people living with HIV (PLHIV) might face a higher burden of hypertension.[4,6, 10-12]”

- 3 Page 5, paragraph 2, last sentence: Please change "...has not been confirmed in this setting" to "has not been replicated in this setting".

Response: We agree that this wording is clearer and have altered the text as suggested.

Introduction

"Just one of these studies examined the effect of antiretroviral therapy (ART) on hypertension, reporting that long-term use of ART was associated with increased prevalence of hypertension, a finding which has not been replicated in this setting"

- 4 Some argument as to the proposed reasoning for there to be a difference in hypertension rates between positive and negative individuals would be beneficial.

Response: We have added argument explaining the mechanisms by which hypertension prevalence may vary between people living with HIV and HIV-negative people.

Introduction

"Various plausible mechanisms that could generate a difference in the prevalence of hypertension between people living with HIV (PLHIV) and HIV-negative people have been proposed. For example, chronic inflammation, increased microbial translocation, renal disease, and blood vessel damage resulting from long-term ART exposure could increase the prevalence of hypertension among PLHIV, as could higher levels of behavioural risk factors among PLHIV in some communities.[4-6] Conversely, possible reasons for a reduced hypertension burden among PLHIV include low blood pressure resulting from advanced HIV disease, better control of blood pressure due to additional healthcare support and lower levels of behavioural risk factors among PLHIV in some settings.[7-9] The conflicting outcomes of the proposed mechanisms mean that it is unclear whether any difference in hypertension prevalence by HIV status exists and in what direction it acts."

Methods

- 5 Analysis would have benefitted from assessing current ART use as well as "ever" use. It is not clear to me what mechanism would be responsible for hypertension in someone who has taken it at one time (perhaps even years ago, perhaps only surrounding labour and delivery for example) but who is not currently on ART.

Response: We agree that it is interesting to explore how current ART usage is associated with hypertension and have introduced an additional investigation exploring this as a sensitivity analysis.

Methods

"As a sensitivity analysis, we explored how the results changed when current ART usage, defined as reporting both having ever taken ART and not having stopped taking ART, was assessed instead of ART-exposure, using the same approach. We also used the same methods to examine the association between adherent current ART usage and hypertension, with adherent current ART usage defined as reporting all the following: having ever taken ART, not having stopped taking ART, taking ART all the time, not forgetting to take ART, and having taken ART every day in the last month."

Results

"Results for sensitivity analyses focused on current ART usage and adherent current ART usage were similar to those for ART-exposure (Supplementary tables 2-4). Fully adjusted multivariable logistic regression models showed no evidence of a relationship between current ART usage and hypertension (0.80, 0.51-1.25, $p=0.312$; Supplementary table 3), or adherent current ART usage and hypertension (0.70, 0.46-1.08, $p=0.107$; Supplementary table 4)."

Results

- 6 Page 15, line 38: Please change “In logistic regression analysis, there was not strong evidence of a relationship....” To “In logistic regression analysis, there was no evidence of a relationship....”

Response: We have altered this text.

Results - Association between ART-exposure and self-reported hypertension

“In multivariable logistic regression analysis, there was no evidence of a relationship between ART-exposure and self-reported hypertension after adjustment for age and sex (OR=0.87, 95% CI=0.56-1.33, p=0.519) or after further adjustment for site (0.83, 0.53-1.30, p=0.411; Table 3).”

- 7 Many comparisons are quoted in this paper, and there is no discussion of correcting for multiple comparisons. Further, it is difficult to differentiate between univariate and multivariate results as they are stated in the text – further adding to the sense that many comparisons were made.

Response: We agree with the reviewer that there were many comparisons quoted in the paper. In response to comments from another reviewer, we have now simplified our results and removed many of these. The results are now focused on the association between HIV status and hypertension, and the association between ART status and hypertension.

To differentiate the univariable results from the multivariable results more clearly, we have included the words “univariable” and “multivariable” in the results section. We have not used the words “univariate” and “multivariate”, as these words refer to the number of dependent or outcome variables, rather than the number of independent variables or confounders included.[4]

We have also added discussion around correcting for multiple comparisons to our text on Strengths and limitations.

4. Hidalgo B, Goodman M. Multivariate or multivariable regression? Am J Public Health 2013;103:39–40. doi:10.2105/AJPH.2012.300897

Results - Association between HIV and self-reported hypertension

“The prevalence of self-reported hypertension in PLHIV (14.1%, 11.9%-16.3%) was similar to the prevalence of self-reported hypertension in HIV-negative people in crude univariable analyses (13.3%, 12.6%-14.0%, F=0.4, p=0.503). The lack of evidence for a difference in univariable analyses remained when analysing the prevalence of self-reported hypertension within age and sex groups for most groups (Fig. 2, Supplementary table 1). However, there was statistical support for lower prevalence of self-reported hypertension in PLHIV than in HIV-negative people among women aged 45-54 (PLHIV: 20.2%, 13.8%-26.7%; HIV-negative: 30.1%, 25.7%-34.5%; F=5.4, p=0.021) and men aged over 65 (PLHIV: 6.7%, 0.0%-19.3%; HIV-negative: 31.2%, 25.3%-37.2%, F=4.1, p=0.043) in univariable analyses.”

“In the initial multivariable logistic regression analysis adjusted for age and sex, no evidence was found for an association between HIV and self-reported hypertension (odds ratio, OR=0.87, 95% CI=0.70-1.07, p=0.191). Introducing further confounders (employment, relationship status, wealth, and site) to generate a fully adjusted multivariable model did not alter this finding (0.88, 0.70-1.10, p=0.261; Table 2).”

Results - Association between ART-exposure and self-reported hypertension

“The prevalence of self-reported hypertension was similar among ART-exposed PLHIV (14.8%, 12.0%-17.7%) and ART-naïve PLHIV in crude univariable comparisons (12.8%, 9.1%-16.4%, F=0.7, p=0.388)

and the lack of a difference remained in univariable analyses of the prevalence of self-reported hypertension split by age and sex (Supplementary fig. 1).”

“In multivariable logistic regression analysis, there was no evidence of a relationship between ART-exposure and self-reported hypertension after adjustment for age and sex (OR=0.87, 95% CI=0.56-1.33, p=0.519) or after further adjustment for site (0.83, 0.53-1.30, p=0.411; Table 3).”

“Fully adjusted multivariable logistic regression models showed no evidence of a relationship between current ART usage and hypertension (0.80, 0.51-1.25, p=0.312; Supplementary table 3), or adherent current ART usage and hypertension (0.70, 0.46-1.08, p=0.107; Supplementary table 4).”

Discussion - Strengths and limitations

“These fall across five broad areas: the definition of hypertension, unmeasured risk factors for hypertension, selection biases towards the healthy, the use of a large number of significance tests, and the cross-sectional nature of the analysis.”

Discussion - Strengths and limitations

“In addition, the analysis involved a large number of significance tests, without correcting for multiple comparisons, which can increase the probability of a false positive finding due to chance.”

Discussion

- 8 Page 18, line 9-13: “A variety of physiological...vital to further understanding”. This would be well-suited to the introduction – in fact – some argument as to the proposed reasoning for there to be a difference in hypertension rates between positive and negative individuals would be beneficial.

Response: We have adapted the sentence and moved it to the introduction, alongside reasoning for there to be a difference in hypertension prevalence between people living with HIV and HIV-negative individuals, which was added in response to comment 4.

Introduction

“As a variety of physiological and treatment-related mechanisms by which HIV could raise or lower hypertension risk have been proposed, epidemiological studies are vital to furthering understanding.[5,7]”

- 9 Page 18, discussion in first paragraph: Another possibility is that those living with HIV may have been more likely to have been told they had hypertension when they went to their physician/clinic to obtain medication or other care for their HIV. Those not living with HIV may have encountered health care providers less frequently and had their blood pressure checked less frequently resulting in an under-reporting of hypertension in this group compared to the group living with HIV. This point needs to be discussed here.

Response: We agree with the reviewer about the possibility that people living with HIV may have been more likely to have been told that they had hypertension, due to their more frequent contact with health services. However, this is discussed in detail in our limitations section, where we have a paragraph focused on understanding the effects of relying on self-reported hypertension, including consideration of differences between people living with HIV and HIV-negative people in their contact with the health system. We feel it would introduce repetition to discuss this again elsewhere in the discussion.

Discussion – Strengths and limitations

“Firstly, funding limitations led to a reliance on self-report of previous diagnoses by a doctor or nurse as a proxy to detect disease, which may have resulted in underestimation of the underlying hypertension

burden.[37] Limited access to screening and diagnosis will likely have meant that some hypertensive people in our study population were at the early stages of the hypertension care cascade and unaware of their hypertension, and so were excluded from our estimates.[39-41] Notably, a 2015 meta-analysis of four studies indicated that the prevalence of hypertension in the general population in Zimbabwe may be 30%, which is higher than our estimates of the prevalence of self-reported hypertension among PLHIV and HIV-negative people. The difference between the two reports suggests that a substantial proportion of the participants in this study may have been unaware that they had hypertension and so were not included in our estimates. [42] This would reduce the generalisability of our findings for settings in which hypertension screening and diagnosis is more readily available. In addition, HIV-negative people may have had less contact with the healthcare system for hypertension diagnosis than PLHIV, which could have artificially reduced self-reporting of hypertension in this group.[43] Yet the prevalence of hypertension was comparable among PLHIV who knew their status (14.7%, 11.9%-17.5%) and PLHIV who did not (15.1%, 10.6%-19.5%), suggesting that contact with the healthcare system among diagnosed PLHIV did not increase self-reporting of hypertension. In future survey rounds, quantification of the prevalence of hypertension by measuring participants' blood pressure during the individual interview, and asking questions to detect medications for hypertension, would provide more reliable estimates of this burden.[44] The results of this analysis could then be updated accordingly."

- 10 Page 18, discussion in second paragraph: This reviewer would argue that newer Integrase inhibitors which have been associated with weight gain could also play a role in hypertension – as increased weight is associated with increased rate of hypertension in the general population. Please amend accordingly.

Response: We agree with the reviewer and have amended the text.

Discussion

"The use of newer integrase inhibitors may also increase weight gain among PLHIV and raise the prevalence of hypertension further. [32,33]"

- 11 Page 19, line 18: NCD – This term is defined at the very beginning of the article, then not used until the end. Please remove and write out in full.

Response: We have altered the text to remove the NCD acronym.

Methods - Data

"The most recent round, which ran between July 2018 and December 2019, is unique as it provides the first insight into the burden of hypertension and other non-communicable diseases."

Discussion - Strengths and limitations

"This study also used robust methodology to explore links between HIV status, ART-exposure, and self-reported hypertension, building the evidence base on links between HIV and non-communicable diseases."

Discussion - Conclusion

"Zimbabwe is re-orienting its health system towards control of hypertension and other non-communicable diseases; this study highlights how carefully designed, sufficiently resourced prevention and care interventions for PLHIV and HIV-negative people will be required to make this a success."

12 Page 19, line 48-52: Measurement of blood pressure, and questions to detect medications for hypertension would also be important.

Response: We have clarified our comments about the need to measure participants blood pressure and highlighted that questions to detect medications for hypertension would also be valuable.

Discussion – Strengths and limitations

“In future survey rounds, quantification of the prevalence of hypertension by measuring participants’ blood pressure during the individual interview, and asking questions to detect medications for hypertension, would provide more reliable estimates of this burden.”

13 Page 20, lines 4-8: BMI is also an important factor affecting hypertension. Height and weight should also be recorded in future studies and analyses adjusted for this.

Response: We agree that BMI plays an important role in hypertension and have adjusted the text to reflect this.

Discussion - Strengths and limitations

“As well as limitations linked to the definition of hypertension, data were not available on several risk factors for hypertension, such as body mass index (BMI), diet and physical activity, which precluded their use in the analysis [45,46]. Future studies should measure the height and weight of participants and calculate their BMI, and gather data on diet and physical activity levels among participants, so that analyses can be adjusted for these risk factors.”

14 Please include discussion of additional limitations to the generalizability of this study:

- a. The Study population is more educated and of higher socioeconomic status than average in the country. The study included only 3% of those from the lowest socioeconomic status and is therefore not generalizable to that group.

Response: We believe that our study cohort is not considerably more educated than would be expected in Zimbabwe, where education levels are higher than in other countries in sub-Saharan Africa.[5] For example, estimates of the proportion of the population who are literate from the Manicaland Cohort approximately align with those from the ZimStat Inter-Censal Demographic Survey 2017.[6] As a result, we have not included this as a specific limitation.

Regarding socio-economic status, our measure of asset-based wealth was not generated relative to the rest of the country. This may not have been clear as we did not provide much detail on how the wealth index was generated. Essentially, the calculation was based on data on ownership of fixed and sellable assets within households. The fixed assets were assets that could not be easily sold, and included the water supply, toilet facilities, electricity supply, housing structure, and floor type of the household. The sellable assets were items that families could easily sell, and included a radio, a television, a bicycle, a motorbike, a car, and one or more cattle. For each item, binary and categorical responses were coded so that they lay between zero and one. For example, owning a bicycle was coded as one, whereas not owning a bicycle was coded as zero. Similarly, having a natural floor of earth, sand or dung was coded as zero, having a rudimentary plank or bamboo floor was coded as 0.5, and having a finished floor of wood or cement was coded as one. The values for each asset owned by a household were summed, and each total was divided by the maximum possible score (eleven, equal to the number of assets) to give an overall asset ownership score between zero (no assets owned) and one (all assets owned) for each household. Each person in the dataset was then given the score for their household. We then split the individuals into five groups, which each covered an equal section of the score scale (0-0.2, >0.2-0.4, >0.4-0.6, >0.6-0.8, and >0.8-1) to create an absolute wealth index for analysis. This means that the small number of people in the lowest asset ownership group reflected the

small number of people from households with very few assets, rather than providing a comparison against the rest of the country.

We accept that using the absolute wealth index could cause confusion and may not have best captured the pattern of wealth across the population, so we have now switched to using a relative wealth index. To create the relative wealth index, the asset ownership scores for each household were calculated in the same way as for the absolute index. Households were then grouped into five equally sized quintiles to create a household level wealth index, with the same number of households in each quintile. Each person in the dataset then received the wealth index associated with their household. While this measure is still not a comparison against the rest of the country, we feel that it will reduce the potential for confusion and better captures the variation in asset ownership, demonstrating that we have not excluded poorer individuals. We have now re-run the analyses with this measure, added additional text to explain how the wealth index is constructed, and updated our findings.

Finally, we would like to highlight that we found that individuals from poorer households were more likely to consent to participate than individuals from wealthier households, suggesting that generalisability of our results to the poorest individuals may exceed generalisability to other groups.

5. Gregson S, Mugurungi O, Eaton J, et al. Documenting and explaining the HIV decline in east Zimbabwe: the Manicaland General Population Cohort. *BMJ Open* 2017;7:e015898. doi:10.1136/bmjopen-2017-015898
6. Zimbabwe National Statistics Agency. Inter-censal Demographic Survey. Harare: 2017. http://www.zimstat.co.zw/wp-content/uploads/publications/Population/population/ICDS_2017.pdf.

Abstract

“Adjusting for socio-demographic variables in logistic regression did not alter this finding (odds ratios: HIV status: 0.88, 0.70-1.10, p=0.261; ART-exposure: 0.83, 0.53-1.30, p=0.411).”

Methods

“Wealth was classified using an index based on whether households owned a range of assets, as described by Lopman et al.[25] Assets included sellable items, such as bicycles and cattle, as well as fixed assets, such as water and electricity supplies. For each asset, binary and categorical responses were coded so that they lay between zero and one. For example, owning a bicycle was coded as one, whereas not owning a bicycle was coded as zero. The values for each asset were then summed, and the total was divided by the overall number of assets, to give an asset ownership score between zero (no assets owned) and one (all assets owned) for each household. Households were then grouped into five equally sized quintiles to create a household level wealth index with five categories. Each person in the dataset was then given the wealth index for their household.”

Results

“More of those who did not consent came from wealthier households ($X^2=47.9$, $p<0.001$)”

Results - Association between HIV and self-reported hypertension

“Introducing further confounders (employment, relationship status, wealth, and site) to generate a fully adjusted multivariable model did not alter this finding (0.88, 0.70-1.10, p=0.261; Table 2).”

Tables and supplementary material have also been updated.

- b. Women had higher odds of self-reported hypertension – could this be related to frequency of visits to a physician rather than a true increase in prevalence?
- c. Wealthiest – higher odds of reporting hypertension than poor – and city vs. subsistence farming area – is this related to access to care (i.e., such that those who are poorer and more rural are less likely to access a care provider to ever be told they have hypertension)?

Response: We agree that it is possible that some of the differences by gender and wealth are the result of more frequent attendance at healthcare facilities. However, we have removed reporting of the odds of hypertension for these confounders from the text, as recommended by another reviewer. The removal of the results for the confounders from the text means that we believe that it would not make sense to add limitations related to these results to the discussion. This is especially the case as we have already included substantial discussion about the limitations of relying on self-reported measures of hypertension.

Discussion – Strengths and limitations

“Firstly, funding limitations led to a reliance on self-report of previous diagnoses by a doctor or nurse as a proxy to detect disease, which may have resulted in underestimation of the underlying hypertension burden.[37] Limited access to screening and diagnosis will likely have meant that some hypertensive people in our study population were at the early stages of the hypertension care cascade and unaware of their hypertension, and so were excluded from our estimates.[39-41] Notably, a 2015 meta-analysis of four studies indicated that the prevalence of hypertension in the general population in Zimbabwe may be 30%, which is higher than our estimates of the prevalence of self-reported hypertension among PLHIV and HIV-negative people. The difference between the two reports suggests that a substantial proportion of the participants in this study may have been unaware that they had hypertension and so were not included in our estimates. [42] This would reduce the generalisability of our findings for settings in which hypertension screening and diagnosis is more readily available. In addition, HIV-negative people may have had less contact with the healthcare system for hypertension diagnosis than PLHIV, which could have artificially reduced self-reporting of hypertension in this group.[43] Yet the prevalence of hypertension was comparable among PLHIV who knew their status (14.7%, 11.9%-17.5%) and PLHIV who did not (15.1%, 10.6%-19.5%), suggesting that contact with the healthcare system among diagnosed PLHIV did not increase self-reporting of hypertension. In future survey rounds, quantification of the prevalence of hypertension by measuring participants’ blood pressure during the individual interview, and asking questions to detect medications for hypertension, would provide more reliable estimates of this burden.[44] The results of this analysis could then be updated accordingly.”

VERSION 2 – REVIEW

REVIEWER	Shamu, Tinei Newlands Clinic
REVIEW RETURNED	26-Nov-2022
GENERAL COMMENTS	All my review comments have been adequately addressed by the authors.
REVIEWER	Murray, Melanie British Columbia Women's Hospital and Health Centre Women's Health Research Institute
REVIEW RETURNED	19-Dec-2022
GENERAL COMMENTS	Thank you for addressing my previously identified concerns.